# video-SALMONN-o1: Reasoning-enhanced Audio-visual Large Language Model

Guangzhi Sun [1 2 3]   Yudong Yang [1]   Jimin Zhuang [1]   Changli Tang [1]   Yixuan Li [1]   Wei Li [2]   Zejun Ma [2]
Chao Zhang [1]

## Abstract

While recent advancements in reasoning optimization have significantly enhanced the capabilities of large language models (LLMs), existing efforts to improve reasoning have been limited to solving mathematical problems and focusing on visual graphical inputs, neglecting broader applications in general video understanding. This paper proposes *video-SALMONN-o1*, the first open-source reasoning-enhanced audio-visual LLM designed for general video understanding tasks. To enhance its reasoning abilities, we develop a reasoning-intensive dataset featuring challenging audio-visual questions with step-by-step solutions. We also propose process direct preference optimization (pDPO), which leverages contrastive step selection to achieve efficient step-level reward modelling tailored for multimodal inputs. Additionally, we introduce RivaBench, the first reasoning-intensive video understanding benchmark, featuring over **4,000** high-quality, expert-curated question-answer pairs across scenarios such as standup comedy, academic presentations, and synthetic video detection. video-SALMONN-o1 achieves **3-8**% accuracy improvements over the LLaVA-OneVision baseline across different video reasoning benchmarks. Besides, pDPO achieves **6-8**% improvements compared to the supervised fine-tuning model on RivaBench. Enhanced reasoning enables video-SALMONN-o1 zero-shot synthetic video detection capabilities.[1]

## 1. Introduction

The recent advancements in optimizing the reasoning process have further boosted text-based large language models (LLMs) (OpenAI, 2024; DeepSeek Team, 2024; Qwen Team, 2024b; Zhao et al., 2024; Yuan et al., 2024) performance in answering complex logical questions, such as math problems (Yang et al., 2024; Wang et al., 2024b; Sun et al., 2024d; Ying et al., 2024) and coding tasks (Zhang et al., 2024f). These methods usually first split the solution into multiple simpler *steps* to form a reasoning path ending with the final solution, as demonstrated in chain-of-thought (CoT) (Wei et al., 2022). Advanced training approaches have been developed such as the outcome reward model (ORM) (Cobbe et al., 2021; Yu et al., 2024a; Zhang et al., 2024b) that optimizes the entire reasoning path based on the final solution, and the process reward model (PRM) (Uesato et al., 2022; Lightman et al., 2023; Luo et al., 2024; Zhang et al., 2024a) that optimizes each reasoning step based on how likely each step would lead to a correct answer.

In addition to text-based questions, reasoning also plays an indispensable role in understanding the physical world, such as comprehending concepts in an academic presentation, interpreting complex interactions among people or even detecting artificial anomalies. Thus, improving reasoning ability is also critical for multimodal LLMs (Tang et al., 2024c;b; Sun et al., 2024b; Cheng et al., 2024; Zhang et al., 2024e; Lin et al., 2024; Team et al., 2024; Wang et al., 2024a; Tang et al., 2024a) that process audio and visual inputs in addition to text, as the interactions among multiple modalities can largely increase the difficulty of the task. To this end, investigations have been performed on optimizing the reasoning process with multimodal inputs (Du et al., 2024), and on particularly visual LLMs (Qwen Team, 2024a; Xu et al., 2024; Du et al., 2025). However, current research on enhancing reasoning capabilities for multimodal LLMs has predominantly focused on solving mathematical problems and image inputs. This overlooks the importance of reasoning in general video understanding and the interactions among audio, visual and text modalities, largely limiting their scopes of applications.

This paper proposes *video-SALMONN-o1*, the first open-source reasoning-enhanced audio-visual LLM with improved reasoning abilities in general video understanding tasks. The audio-visual reasoning capability of video-SALMONN-o1 is first enhanced by creating a new dataset with challenging questions and step-by-step solutions for su-

---

[1]Tsinghua University [2]ByteDance [3]University of Cambridge. Correspondence to: Chao Zhang <cz277@tsinghua.edu.cn>.

*Proceedings of the 42nd International Conference on Machine Learning*, Vancouver, Canada. PMLR 267, 2025. Copyright 2025 by the author(s).

[1]Code available: https://github.com/BriansIDP/video-SALMONN-o1.

pervised fine-tuning (SFT), and then further boosted by the proposed variant of direct preference optimization (DPO), process DPO (pDPO) (Rafailov et al., 2024; Zhang et al., 2024c). pDPO achieves step-level pairwise reward modelling via an efficient contrastive step selection approach tailored for multimodal inputs. While being more effective than the standard PRMs in general video understanding, pDPO and the step selection make audio-visual reasoning more efficient without the need for an external reward model or a two-pass re-ranking pipeline.

To evaluate the performance on multimodal reasoning for general video understanding, we propose the first **r**easoning-**i**ntensive **v**ideo with **a**udio understanding benchmark (RivaBench). RivaBench primarily focuses on three representative scenarios, including standup comedy, academic presentation and synthetic video detection. In particular, RivaBench contains over 4k high-quality question-answer pairs that are carefully crafted by human experts (*e.g.* medical doctors). Our key contributions are summarized as follows:

- We propose video-SALMONN-o1, the first open-source reasoning-enhanced audio-visual LLM for general video understanding tasks.

- video-SALMONN-o1 is the first to explore RL-based reasoning optimization for general video understanding. The proposed pDPO method with efficient contrastive step selection further enhances reasoning abilities.

- We propose RivaBench, the first general video understanding benchmark focusing on challenging audio-visual reasoning scenarios with human expert annotations.

- video-SALMONN-o1 consistently outperforms the strong LLaVA-OneVision visual baseline on VideoMME, NExT-QA and RivaBench, with **3-8**% absolute accuracy improvements. The pDPO training achieved **6-8**% improvements on RivaBench over the SFT model. Moreover, video-SALMONN-o1 is also the first open-source model that showed zero-shot synthetic video detection ability.

## 2. Related Work

### 2.1. CoT Reasoning

CoT reasoning is one of the remarkable abilities of LLMs when solving difficult and complex problems. Earlier investigations employed prompt tuning and various search algorithms such as the Monte-Carlo tree search during inference time (Hao et al., 2023; Snell et al., 2024; Feng et al., 2024; Yao et al., 2023; Goyal et al., 2024). Later on, training stage approaches using reinforcement learning (RL) were developed to further and more radically boost the reasoning capabilities of LLMs. RL has also been used in multimodal LLMs but not from the perspective of improving reasoning (Yu et al., 2024b; Zhang et al., 2024d). PRMs which estimate the value function of each reasoning step have emerged

as one of the most prevalent approaches in reasoning optimization tasks (Uesato et al., 2022; Lightman et al., 2023; Luo et al., 2024; Zhang et al., 2024a; Li et al., 2023b).

However, constructing step-level annotations for PRM training can be expensive and difficult to scale up. As mitigation, Wang et al. (2024b) and Luo et al. (2024) proposed automatic step annotation using *rollout*, which approximated the expected correctness of each step by sampling multiple paths till the end with the same prefix solution. In particular, Luo et al. (2024) treats the first wrong step as the critical step to perform rollout which was found by binary search.

### 2.2. Reasoning in Multimodal LLMs

Researchers have been investigating optimizing CoT reasoning for multimodal LLMs to tackle increasingly challenging tasks. Most of them focus on extracting graphical or text information from an image and solving mathematical tasks based on the extracted information. Specifically, LLaVA-CoT (Xu et al., 2024) investigated better sampling and search algorithms to find a better reasoning path for math questions with image inputs. Virgo, on the other hand, explores the fine-tuning data organization and transferability of text-based reasoning tasks to image-based reasoning tasks (Du et al., 2025). Recently, MAmmoTH-VL (Guo et al., 2024) built a large-scale multimodal instruction-tuning dataset that can improve the question-answering performance on diverse modalities including video. Different from these works, video-SALMONN-o1 particularly focuses on general video understanding scenarios, where different parts of the audio-visual information are constantly referred to during the reasoning process.

### 2.3. Benchmarks for Audio-visual LLMs

The fast-paced development of multimodal LLMs has boosted the creation of more challenging video understanding benchmarks. Benchmark focus evolves from video description and perception abilities (Li et al., 2021; Alamri et al., 2019; Chen et al., 2023a; Li et al., 2022; Chen et al., 2023b; Ning et al., 2023; Mangalam et al., 2023; Yun et al., 2021; Sun et al., 2024a), to video reasoning abilities such as inference about temporal and causal relations (Xiao et al., 2021; Li et al., 2024b; 2023a; Fu et al., 2024; Liu et al., 2024a; Fang et al., 2024). In particular, NExT-QA (Xiao et al., 2021) focuses on causal relation reasoning such as why a certain action is performed, and Video-MME (Fu et al., 2024) contains questions that require the combination of both audio and visual information to perform reasoning. Our proposed RivaBench has more challenging questions that require *longer* thinking steps, *broader* world knowledge and a *tighter* combination of audio-visual information.

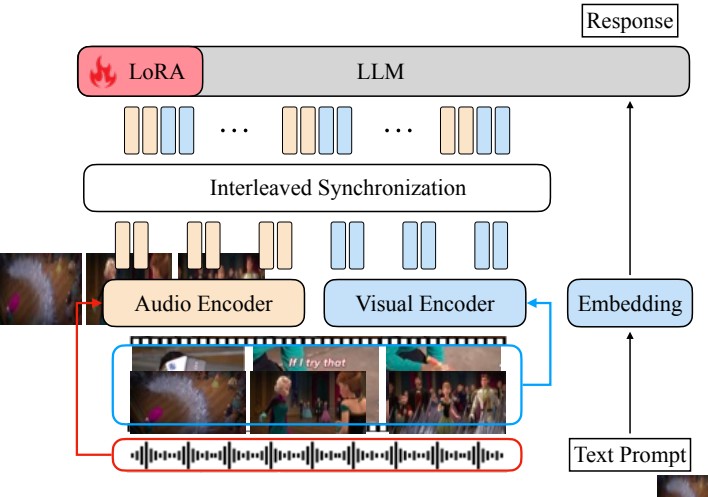

*Figure 1.* video-SALMONN-o1 model structure. The input video is processed by the visual and audio branches, generating encodings from the visual and audio frame sequences respectively. Two encoding streams are combined in an interleaved fashion to synchronize across time before sending to LLM.

# 3. video-SALMONN-o1

## 3.1. Model Structure

We adopt the same model structure as video-SALMONN 2, as shown in Fig. 1. As video-SALMONN 2 (Tang et al., 2024a), the model is built based on a pre-trained visual LLM by adding the audio encoder branch. The input video and audio streams are processed separately by the audio encoder and visual encoder and are then separately mapped to the dimension of the LLM input via individual modality aligners. To combine the audio and visual encodings, the interleaved synchronization module is employed as illustrated in Fig. 1. The groups of encodings per visual frame are equally spaced across time, and the audio encodings corresponding to the time between two visual frames $t_1$ and $t_2$ are inserted between the two groups of visual encodings. The process is summarized as in Eqn. (1):

$$\mathbf{H}^{\text{AV}} = \text{Concat}(\dots, \mathbf{H}^V_{t_1}, \mathbf{H}^A_{t_1:t_2}, \mathbf{H}^V_{t_2}, \dots) \quad (1)$$

where $\mathbf{H}^A \in \mathbb{R}^{m \times d}$ and $\mathbf{H}^V \in \mathbb{R}^{n \times d}$ represent groups of audio and visual encodings, and $m$ and $n$ are the number of encodings in each group.

A multi-stage SFT pipeline with the cross-entropy loss on reference response is adopted to train video-SALMONN-o1 before optimizing the reasoning process with RL. Starting from the pre-trained visual model, the audio aligner is trained from scratch keeping other parts of the model frozen. Then, using paired audio-video data, the modality aligners and the low-rank adaptation (LoRA) module (Hu et al., 2022) are trained with other parts frozen.

## 3.2. Reasoning-intensive SFT Data

We empirically discovered that video understanding models (Cheng et al., 2024; Zhang et al., 2024e; Lin et al., 2024) generally lose the ability to perform step-by-step reasoning when a video is given, and always directly generate the final answer. To re-obtain the reasoning ability during the SFT stage, we create a set of more challenging question-answering pairs based on the same training set videos using proprietary LLMs, and the pipeline is shown in Fig. 2.

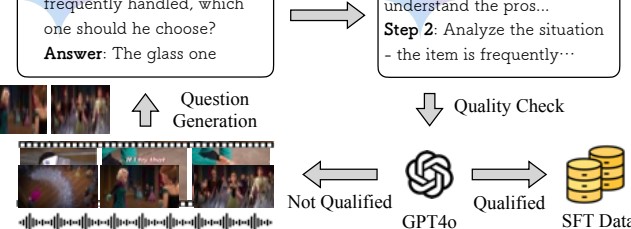

*Figure 2.* Acquisition pipeline of reasoning-intensive SFT data. The question, answer and reasoning paths are generated by Gemini-1.5-pro taking the video with paired audio as inputs. GPT4o is employed for quality checks to ensure the QA-pair and the reasoning steps are valid and require logical thinking.

For each video with paired audio, we use Gemini-1.5-pro to generate a question-answer pair with the reasoning steps. Then, to avoid bias in Gemini models and ensure the quality of the questions and reasoning steps, a quality check stage is employed using GPT-4o. Questions with poor quality will be discarded and a new question-answer pair will be generated again. In addition to the newly created question, we augment the original training set by generating reasoning paths with Gemini-1.5-pro and checking by GPT-4o following the pipeline to avoid network learning two distinct mechanisms for reasoning and direct answer. This turned out to be important to yield competitive reasoning performance from SFT in our empirical study.

# 4. Training to Enhance Reasoning Abilities

## 4.1. Preliminary

The reasoning process refers to the LLM generating the $Q \to \{s_1, s_2, ..., s_K\} \to A$ sequence, where $Q$ is the question, $A$ is the answer and $s_k$ are reasoning steps that logically connect the question $Q$ to the final answer $A$. By treating this as a Markov decision process (MDP) and the LLM as the policy model, PRM is to provide feedback for each step $s_k$ that guides the LLM in making accurate reasoning by optimising the policy to maximise the reward.

Following Wang et al. (2024b), the PRM is to estimate the

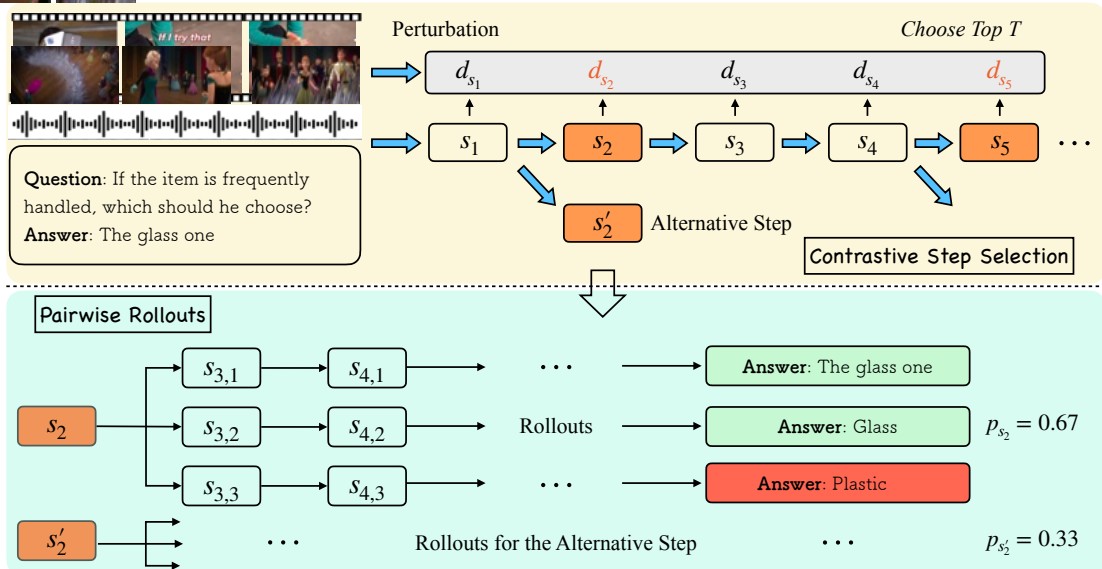

*Figure 3.* Illustration of the contrastive step selection (top) and pairwise rollout (bottom) to construct per-step expected correctness score for pDPO. Contrastive step selection: Top 2 steps, $s_2$ and $s_5$ are selected in this example, and for $s_2$, an alternative step, $s_2'$, is sampled to form the preference pair. Pairwise rollout: Three rollouts are shown for each step and $s_2$ and $s_2'$ are step pairs with the same prefix solution. The answer correctness is checked using GPT-4o by comparing it against the reference answer.

expected answer correctness, $p_{s_k}$, of a prefix solution $\{s_{1:k}\}$. The expected correctness score can be approximated with Monte Carlo sampling of multiple paths from the prefix solution to an answer $A_n$ as shown in Eqn (2).

$$p_{s_k} \approx \frac{1}{N} \sum_{n=1}^{N} \mathbb{1}(A_n = A_{\text{ref}}) \qquad (2)$$

where $A_{\text{ref}}$ is the reference answer and $A_n$ is one sampled answer. The sampled path $\{s_{k+1,n}, s_{k+2,n}, \ldots, s_{K(n),n}\}$ that leads to $A_n$ is referred to as a *rollout*. The PRM training loss is then shown as

$$\mathcal{L}_{\text{PRM}} = \sum_{k=1}^{K} p_{s_k} \log r_{s_k} + (1 - p_{s_k}) \log(1 - r_{s_k}) \quad (3)$$

where $r_{s_k} \in [0, 1]$ is the PRM prediction which can be derived from the LLM output at the last token of each step with a fully connected layer with a sigmoid function $\sigma(\cdot)$.

### 4.2. Process DPO

As pointed out by Zhang et al. (2024a), predicting an absolute score fails to exploit the instruction-following capabilities of LLMs as well as influenced by ambiguities in score standards. Both problems are more severe in audio-visual LLMs. Therefore, we propose pDPO for video-SALMONN-o1, which is a pairwise preference modelling approach by training the model to select the better reasoning path rather than giving absolute scores to the paths. Different from the pairwise preference reward model (PPRM) in (Zhang et al., 2024a) that leverages the partial ordering of entire reasoning

paths, pDPO models the preference for a specific reasoning step given the same prefix solution. Specifically, the reward function for each step of interest can be written as

$$r(s_k) = \beta \log \frac{\pi_\theta(s_k|s_{<k}, \mathbf{H}^{\text{AV}})}{\pi_{\text{ref}}(s_k|s_{<k}, \mathbf{H}^{\text{AV}})} + \beta \log Z(s_{<k}, \mathbf{H}^{\text{AV}}) \quad (4)$$

where $\pi_\theta$, $\pi_{\text{ref}}$, $\beta$ and $Z(\cdot)$ are the LLM policy, reference policy, a parameter controlling the deviation from $\pi_{\text{ref}}$, and the partition function as in Rafailov et al. (2024) respectively. $Z(s_{<k}, \mathbf{H}^{\text{AV}}) = \sum_{s_k} \pi_{\text{ref}}(s_k|s_{<k}, \mathbf{H}^{\text{AV}}) \exp(\frac{1}{\beta} r(s_k))$. For each step, an alternative step $s_k'$ is generated, and pairwise rollout is performed for both steps as shown in Fig. 3. The probability of $s_k$ being better than $s_k'$ is then defined using the Bradley-Terry model as

$$p(s_k \succ s_k') = \sigma(r(s_k) - r(s_k')). \qquad (5)$$

Then, the pDPO loss can be written as:

$$\mathcal{L} = -\mathbb{E}\Big[\alpha_k \log p(s_k \succ s_k') + (1 - \alpha_k) \log p(s_k' \succ s_k)\Big] \quad (6)$$

where $\alpha_k = \mathbb{1}(p_{s_k} > p_{s_k'})$. Alternatively, $\alpha_k = \sigma((p_{s_k} - p_{s_k'})/\mu)$ can be used as soft labels for DPO to accommodate the estimation noise introduced by the limited number of rollouts in $p_{s_k}$, where $\mu$ is the calibration hyper-parameter determining how much we believe the process annotations. As a result, pDPO retains the advantages of PPRM while offering finer modelling granularity at each step. In practice, pDPO is integrated with PPRM to construct complete reasoning paths, enhancing overall performance. While PPRM

enables full-solution-level preference training, ensuring the generation of entire solutions, pDPO complements it by providing fine-grained, step-level preference guidance.

### 4.3. Contrastive Step Selection

While rollouts allow automatic process annotation, the computational cost can be high when the numbers of rollouts and steps grow. However, in pDPO, certain steps are more error-prone and hence more valuable to be optimized than others. For general video understanding, by examining a held-out validation set for reasoning paths with wrong answers, we found that over 70% of the reasoning errors occur at steps where the model misinterprets or hallucinates the video content. Therefore, we choose to particularly focus the pDPO on optimizing those steps.

To locate those steps, we quantify the susceptibility of each reasoning step to the input video by applying a tiny perturbation to the input video and measuring the length-normalized per-token KL divergence. Specifically, as shown in the top part of Fig. 3, for each step $s_k$ we compute the length-normalized KL-divergence by

$$d_{s_k} = \frac{1}{|s_k|} \sum_{y_i \in s_k} D_{\mathrm{KL}}\Big(P(y_i|y_{<i}, \mathbf{H}^{\mathrm{AV}})||P(y_i|y_{<i}, \tilde{\mathbf{H}}^{\mathrm{AV}})\Big),$$

where $y_i$ are wordpiece tokens and $D_{\mathrm{KL}}(\cdot)$ computes the KL-divergence between the output distributions with the original inputs $\mathbf{H}^{\mathrm{AV}}$ and perturbed inputs $\tilde{\mathbf{H}}^{\mathrm{AV}}$. A higher $d_{s_k}$ indicates that the reasoning step $s_k$ is more susceptible to small input change, and this high susceptibility is likely to yield more diverged subsequent steps. We select the top $T$ steps with the highest $d_{s_k}$ to perform pairwise rollout. While this selection biases pDPO training towards video-dependent errors, the other text-based logic errors can be accommodated by PPRM with entire reasoning paths.

## 5. Audio-visual Reasoning Benchmark

The RivaBench is proposed to extend the scope of complex video understanding with three new reasoning-intensive application scenarios, including academic presentation (Academic), stand-up comedy (StandUp) and synthetic video detection (SynthDec). The statistics of videos for each scenario partition are shown in Table 1.

The **Academic partition** is based on the M3AV (Chen et al., 2024) test set containing recordings of conference or lecture presentations spanning five different domains. Human experts with mathematical, engineering and medical backgrounds are recruited to provide questions, answers and detailed explanations based on the video clips. Example annotations are shown in Figs. 9 and 10 in Appendix C.

While humour in videos has been explored from a descriptive perspective (Hyun et al., 2024; Liu et al., 2024b; Xie

*Table 1.* RivaBench basic statistics. The duration is given by mean ± standard deviation. The SynthDec split contains 100 synthetic videos and 100 real videos that human annotators search to have similar content as synthetic videos. MCQ stands for multiple-choice questions. Video sources are all from YouTube.

| Attribute | Academic | StandUp | SynthDec |
|---|---|---|---|
| Num. of QA | 1,912 | 2,128 | 200 |
| Duration (s) | 47.2± 66.1 | 43.2± 15.1 | 8.1±3.2 |
| Format | 5-way MCQ | 5-way MCQ | Yes/No |

et al., 2024), the **StandUp partition** of RivaBench explores from an audio-visual reasoning perspective. Specifically, instead of prompting the model to list all funny elements in the video, we particularly focus on understanding why a certain punchline is interesting and task the human annotators to set questions that require reasoning about the comedian's gestures, facial expression and speech content. Human annotators provide questions, answers and explanations (with automatically generated confusing choices), as shown in Figs. 7 and 8 in Appendix B. Subtitles exist in around 50% of the videos in the StandUp partition but not in the Academic partition. However, due to the low frame sampling rate and hence subtitles are mostly incomplete, the model has to leverage both audio and visual information. This is reflected by the fact that GPT-4o (visual only) is worse than Gemini-1.5-pro (audio-visual) in Table 2.

This paper proposes the **SynthDec partition** for synthetic video detection, which has great potential since video generation models are becoming increasingly powerful. This task requires LLM to classify whether a given video clip is real or synthetic by finding clues in the video such as motions violating physics rules or objects being distorted. Videos are generated using the Hunyuan-large model (Sun et al., 2024c) (see examples in Figs. 11 and 12) This is a challenging task that requires both logical reasoning and accurate perception of video content. The SynthDec partition can also serve as the performance indicator for reward models used to train video generators in the future.

## 6. Experimental Setup

### 6.1. Model and Training Specifications

video-SALMONN-o1 is built based on the SigLIP (Zhai et al., 2023) visual encoder and Qwen 2 with 7B parameters backbone LLM. Two linear layers with GELU activation function are used (Hendrycks & Gimpel, 2016) as the visual aligner. The model processes videos at a 2-frame-per-second rate with a maximum of 60 frames.

The Whisper-Large-v3 encoder (Radford et al., 2023) is used as the audio encoder, and the window-level Q-Former (Tang et al., 2024c) with a window length of 0.2 seconds is

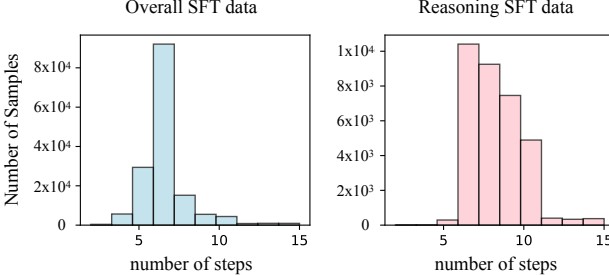

*Figure 4.* Distributions of the numbers of reasoning steps in SFT data. Left: Distribution of the entire SFT data. Right: Distribution on the reasoning-intensive subset of SFT data. Due to the difficulty of the reasoning-intensive subset, more reasoning steps are required in general for samples in this set.

used as the audio aligner, producing 150 audio tokens for every 30 seconds. We set LoRA hyper-parameters $r = 64$ and $\alpha = 256$ for the backbone LLM for both SFT and pDPO. During training, the visual encoder and aligner, audio encoder, and LLM remain frozen. SFT is performed on $16 \times$A100 GPUs for 48 hours and pDPO is trained with $8 \times$A100 GPUs for 24 hours. Prompts used for reasoning are shown in Appendix E. The code, SFT data, pDPO data and model checkpoints will be released.

### 6.2. Data

Following Tang et al. (2024a), the audio modality alignment stage employs LibriSpeech-960h (Panayotov et al., 2015) ASR data and AudioCaps (Kim et al., 2019) audio caption data to train the audio aligner. During the audio-visual SFT stage, 13k videos with rich audio information are selected with high-quality audio-visual captions. Around 150k normal question-answer (QA) pairs are directly generated using GPT-4o by providing detailed audio-visual captions, and an additional subset of 30k reasoning-intensive SFT QA pairs are generated with the proposed data generation pipeline. Each QA, regardless of the difficulty, is associated with reasoning steps, and the distributions of the numbers of reasoning steps for the QA pairs used for SFT are shown in Fig. 4. Both captions and QA pairs are used for SFT.

The reasoning-intensive subset is used to collect the data for pDPO training by sampling 10 paths for each QA. The QA pairs where the SFT model generates incorrect solutions are retained to perform rollouts and others that only contain correct solutions are discarded. For complete solutions, instead of directly comparing the paths (Zhang et al., 2024a), we compare each pair of solutions against the reference answer using GPT-4o and choose the one closer to the reference as the preferred solution. For intermediate steps, we choose the top 3 steps based on contrastive step selection, and 6 rollouts are performed for each chosen step. As a result, ∼100k pairs of complete solutions from 5k video clips are

selected, and an extra 100k pairs of step-level partial solution pairs from these complete solutions are used for pDPO. Greedy decoding is used during inference and the effort in generating data using pDPO only affects training time and has no impact to the latency of the model during inference.

Besides RivaBench, video-SALMONN-o1 is also evaluated on Video-MME (Fu et al., 2024) and NExT-QA (Xiao et al., 2021) benchmarks with challenging reasoning questions where the former is an audio-visual task and the latter focuses on visual information only. For consistency, paired audios are also provided for NExT-QA videos if they exist. Note that the synthetic video detection task is **never seen** in model training, and hence is a zero-shot emergent ability.

## 7. Results

### 7.1. Main Results

The main results on VideoMME, NExT-QA and the RivaBench are shown in Table 2. No subtitles are given to any of the models under test for VideoMME. As performance references, we include GPT-4o (checkpoint at 2024-08-06) and Gemini-1.5-pro, with their results on VideoMME as reported in Fu et al. (2024). When testing GPT-4o with videos, each video is split into images at a frame rate of 2 fps with a maximum of 30 frames due to token limitation, and the sequence of images is sent as the input. For open-source models, we compare video-SALMONN-o1 to LLaVA-OneVision (Li et al., 2024a) (same visual encoder and LLM backbone), together with video-SALMONN (Sun et al., 2024b) and Video-LLaMA 2 (Cheng et al., 2024) as the two most recent audio-visual LLMs.

**Proprietary LLM performance on RivaBench**: For the two proprietary LLMs, GPT-4o underperforms Gemini-1.5-pro on StandUp and Academic test sets due to the lack of audio information. This indicates that RivaBench provides challenging questions that require more audio-visual joint understanding compared to VideoMME. On the SynthDec set, since only the visual part is synthesized, GPT-4o demonstrated a stronger ability. Moreover, by performing reasoning with GPT-4o and Gemini-1.5-pro, larger improvements are found on StandUp and Academic test sets than VideoMME and NExT-QA, indicating the necessity of reasoning on RivaBench.

**Open-source LLM performance comparison**: Audio-visual SFT on video-SALMONN-o1 already yields better performance than LLaVA-OneVision on VideoMME due to the ability to comprehend speech and audio information, whereas no obvious improvements are found on the other benchmarks. The main improvements on other benchmarks come from pDPO, which achieved 4.1%, 8.1% and 5.8% absolute accuracy improvements on NExT-QA, StandUp and Academic test sets respectively compared to the SFT

*Table 2.* Main results of video-SALMONN-o1 compared against other visual (V) and audio-visual (A+V) LLMs. SFT refers to the model after SFT with reasoning data and pDPO refers to the model obtained after training with pDPO based on the same SFT model. F1-score (Precision/Recall) is reported for SynthDec and accuracy is reported for others. Results with † are directly taken from the corresponding papers. video-SALMONN-o1 performs reasoning during inference and other open-source models give answers directly.

| Model | Modality | VideoMME | NExT-QA | RivaBench | | |
| --- | --- | --- | --- | --- | --- | --- |
| | | | | StandUp | Academic | SynthDec (P/R) |
| *Proprietary models* | | | | | | |
| Gemini-1.5-pro (Team et al., 2024) | A+V | 75.0%† | 79.2% | 75.8% | 67.1% | 23.6% (55%/15%) |
| Gemini-1.5-pro+reasoning | A+V | 75.1% | 79.5% | 81.8% | 69.5% | 40.0% (49%/34%) |
| GPT-4o (OpenAI Team, 2024) | V | 71.9%† | 81.7% | 63.3% | 60.0% | 34.1%(90%/21%) |
| GPT-4o+reasoning | V | 72.1% | 81.9% | 69.6% | 61.0% | 25.8%(53%/17%) |
| *Open-source baselines* | | | | | | |
| LLaVA-OneVision (Li et al., 2024a) | V | 58.2%† | 79.4%† | 67.2% | 45.8% | 0.0%(97%/0%) |
| video-SALMONN (Sun et al., 2024b) | A+V | 43.3% | 49.2% | 47.8% | 33.6% | 0.0%(100%/0%) |
| Video-LLaMA 2.1 (Cheng et al., 2024) | A+V | 54.9%† | 75.6% | 53.7% | 34.3% | 0.0%(99%/0%) |
| video-SALMONN-o1 (ours, SFT) | A+V | 62.9% | 78.2% | 68.6% | 42.5% | 5.8%(97%/5%) |
| video-SALMONN-o1 (ours, pDPO) | A+V | **65.6%** | **82.3%** | **76.7%** | **48.3%** | **17.8%**(87%/13%) |

*Table 3.* Effect of different parts of the audio-visual SFT data on VideoMME, Academic and StandUp test sets. Underscore for second-best results. "w/o reasoning-intensive part" means removing the reasoning-intensive SFT data, and "w/o any reasoning" always directly outputting answers during SFT. "Reasoning-intensive part only" always performs reasoning for QA.

| Training Data | Inference Reasoning | VideoMME | NExT-QA | Academic | StandUp |
| --- | --- | --- | --- | --- | --- |
| Full SFT data | ✗ | 63.7% | 80.7% | 45.2% | 72.3% |
| Full SFT data | ✓ | 62.9% | 78.2% | 42.5% | 68.6% |
| w/o any reasoning | ✗ | 63.2% | 81.0% | 44.1% | 71.1% |
| w/o reasoning-intensive part | ✗ | 62.7% | 78.9% | 44.7% | 71.5% |
| w/o reasoning-intensive part | ✓ | 61.6% | 76.6% | 42.3% | 67.5% |
| Reasoning-intensive part only | ✓ | 58.8% | 75.2% | 40.1% | 63.5% |
| Full SFT data + pDPO | ✓ | **65.6%** | **82.3%** | **48.3%** | **76.7%** |

model. Larger improvements are found on the RivaBench with 6-8% absolute accuracy improvements obtained compared to LLaVA-OneVision, and video-SALMONN-o1 even performs better on the StandUp test set than Gemini-1.5-pro without reasoning. Besides, compared to other audio-visual LLMs, video-SALMONN-o1 exhibits better interpretability of the model output, and the cause of mistakes can be located by analyzing the reasoning process. We also provide the comparison between video-SALMONN-o1 and Gemini-1.5-pro on VideoHallucer benchmark (Wang et al., 2024c) in Appendix I for the hallucination aspect.

**Zero-shot synthetic video detection**: video-SALMONN-o1 achieves zero-shot synthetic video detection ability while other open-source models output "real" all the time, which also benefit from a better explanation with examples of anomalies in synthesized videos in the prompt. However, even for the videos where the motions obviously violate physics rules, current state-of-the-art video LLMs still fail

to detect most of the time.

In addition, two qualitative examples are shown in Figs. 16 and 17 in Appendix G, where LlaVA-OneVison (and also other audio-visual models) are unable to provide the reasoning steps and the final answer is completely biased to "Real". On the other hand, video-SALMONN-o1 can look for distortions in the video as part of its reasoning process, leading to the correct identification of synthesized videos.

### 7.2. Effect of SFT Data

The audio-visual SFT data is crucial for video-SALMONN-o1 to gain the initial audio-visual reasoning ability, and the effect of different data partitions is shown in Table 3.

**Direct answer outperforms reasoning after SFT**: Directly outputting a short answer or an option has been the dominating output mode for audio-visual LLMs on general video understanding, a major difference to math questions. Com-

*Table 4.* Effect of different reward modelling methods on VideoMME, NExT-QA, the StandUp and Academic split of RivaBench. Major@20 and RM@20 are evaluated following Zhang et al. (2024a), where Major@20 refers to the accuracy under majority voting with 20 sampled paths, and RM@20 is the best-of-n with 20 samples. Samples are all generated from the model after SFT. pDPO with full paths only uses preference pairs of complete reasoning paths.

| Training Configuration | Inference | VideoMME | NExT-QA | StandUp | Academic |
| --- | --- | --- | --- | --- | --- |
| SFT | 1-best | 62.9% | 78.2% | 68.6% | 42.5% |
| SFT | Major@20 | 63.5% | 81.5% | 73.5% | 45.3% |
| SFT + ORM | RM@20 | 62.7% | 78.5% | 69.0% | 42.6% |
| SFT + PRM | RM@20 | 63.5% | 79.3% | 72.1% | 43.9% |
| SFT + pDPO | 1-best | **65.6%** | **82.3%** | **76.7%** | **48.3%** |

paring row 2 to row 1 in Table 3, when using all the SFT data including the reasoning-intensive part, the model after SFT is still better at directly generating the answer than performing reasoning. This is due to the exposure bias in teacher forcing which has a much higher impact on the reasoning paths as they are much longer sequences. By learning on its own samples, pDPO mitigates this exposure bias and achieves consistently better performance than the SFT model. Next, comparing row 1 to row 3 and row 4 in Table 3, when directly outputting the answer during inference, incorporating reasoning steps in SFT does not always yield an improvement on videoMME and NExT-QA, despite being slightly helpful on RivaBench.

**The reasoning-intensive part is important**: When excluding the reasoning-intensive part, there is a clear degradation in model performance with reasoning during inference, showing the importance of this part of data to enable a better reasoning performance. However, when only using the reasoning-intensive part for SFT, the model struggles to acquire the fundamental audio-visual perception abilities, yielding sub-optimal performance.

### 7.3. Effect of pDPO Training

We then analyse different reward modelling techniques for the model performance in Table 4. In addition to pairwise preference models, we include ORM and PRM as proposed in Lightman et al. (2023) as follows:

**ORM**: A projection layer is added to LLM output states and projects the last output state to a scalar. which is then passed through a sigmoid activation function to predict 1 if the final answer is correct, and 0 otherwise.

**PRM**: A projection layer is added to LLM output states and projects the state at the end of each step to a scalar with a sigmoid to predict $\mathbb{1}(p_{s_k} > 0)$. The score of each solution is the lowest score among all steps (Wang et al., 2024b).

Both ORM and PRM are initialized with video-SALMONN-o1 after the SFT stage. Best-of-n is used for ORM and PRM where 20 sampled solutions are generated from the SFT model and the top one with the highest score is selected.

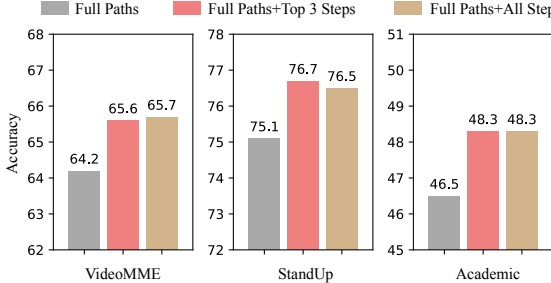

*Figure 5.* Comparison between different top T steps selected for pDPO. Pairs of full solution paths are always used in addition to pairs of intermediate steps.

Moreover, majority voting among the 20 samples is used as a baseline which is consistently marginally better than the 1-best solution across all test sets.

While ORM showed mixed results compared to the 1-best solution from the SFT model, PRM showed consistent but marginal improvements and is on par with majority voting. The training loss of PRM and ORM only dropped about 5%, which reflects the difficulty of learning raw scores for general video understanding tasks. Last, comparing the models above against the pDPO model, the use of pairwise preference models is much more effective compared to predicting the raw score, showing the difficulty of direct raw score modelling in general video QAs. Qualitative examples comparing answers between SFT and pDPO are provided in Figs. 13 to 15 in Appendix F.

**Effect of Contrastive Step Selection**. To analyze the effect of the number of steps selected for pairwise training, we conducted experiments without intermediate pairs of steps and with all intermediate pairs, in addition to using the top three steps from the contrastive step selection. The comparisons are given in Fig. 5. Using intermediate steps in pDPO achieved further consistently improves model performance compared to only using the full solutions, especially on questions that require frequent reference to the video or audio information at intermediate reasoning steps. A case study qualitatively showing the effect of contrastive step selection is included in Appendix H.

## 8. Conclusions

We propose video-SALMONN-o1, the first open-source audio-visual LLM with enhanced reasoning abilities. video-SALMONN-o1 is the first to explore reasoning process optimization for general video understanding and proposes the pDPO method with an efficient contrastive step selection algorithm. To further evaluate the reasoning abilities of audio-visual LLMs, the RivaBench is introduced with innovative and challenging tasks and over 4000 high-quality human expert annotations. video-SALMONN-o1 consistently outperforms the strong LLaVA-OneVision baseline with 3-8% absolute accuracy improvements. pDPO training consistently outperformed the SFT model. Moreover, video-SALMONN-o1 showed zero-shot synthetic video detection abilities as a result of the enhanced reasoning abilities.

## Impact Statement

By enhancing reasoning abilities in general video understanding, video-SALMONN-o1 provides a more transparent and interpretable interface that is compatible with general videos to access and explain model responses and behaviours. This is indispensable to ensure the reliability of LLMs when applied to different video understanding scenarios and will be largely beneficial for pinpointing the specific causes or errors when the model generates dubious or toxic contents, thus enhancing AI safety.

The approaches in this paper do not give rise to any additional potential biases beyond the ones directly inherited from the pre-trained model checkpoints used. The audio encoder and visual encoder might work worse for people from particular demographics. The framework also inherits biases from all the LLMs used in this paper. To mitigate potential biases, we clearly describe the nature of each dataset and provide clear and adequate references to all the resources we used for video-SALMONN-o1.

The ability of video-SALMONN-o1 to understand speech in videos could lead to potential technology abuses like surveillance and eavesdropping. To counter this, we've consulted with legal experts to establish clear usage guidelines, reducing risks and addressing concerns, highlighting our dedication to responsible research sharing.

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

## A. Reasoning SFT Data Example

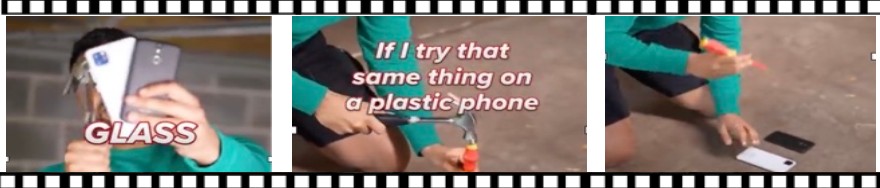

**Speech:** 99% of smart phones are either made of glass or plastic, but which one's better…

**Question**: If the item is frequently handled but almost no dropping risk, which type of material should he choose?

**Answer**: The glass one

**Reasoning**:

1. The question focuses on two aspects: (1) frequent handling, which implies a higher risk of scratches, and (2) minimal dropping risks, suggesting...
2. Analyze the first test (scratch test): ...
3. Analyze the second test (dropping test): ...
4. Compare the two test under the given scenario: ...
5. Conclude that under the given scenario, the glass one is preferred
6. The final answer: the glass one

*Figure 6.* Example of reasoning SFT data

## B. StandUp Data Examples

Two examples of the StandUp part of RivaBench are shown in Fig. 7 and 8 respectively.

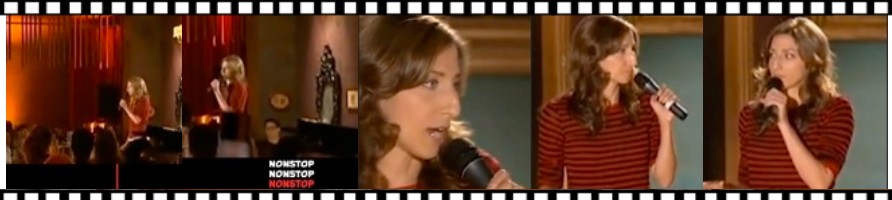

**Speech and audio:** Thank you guys. Are you guys trying to judging my attractive level? It's weird level. Frustrating, and I find that out from things people say to me like, my friend's mom, like, "Chelsea! I saw an ugly Chelsea on the street the other day!". Basically that translate to me like you saw an ugly person on the street and you thought about me. [audience laughed]

**Question**: Why did the audience laugh when the comedian mentioned her friend mom saying she saw an ugly person on the street?

**Answer**: She is using self-deprecating humor to make a funny situation.

**Explain**: The audience laughs because the person uses self-deprecating humor to make a funny situation. By sharing the comment from their friend's mom, who saw an ugly person on the street and thought of her, the person is employing self-mockery.

**Choices**:

A. A self-deprecating way to make a funny situation
B. The comedian is known for having many identical-looking friends
C. The comedian's outfit is considered ugly
D. The street the comedian's friend's mom mentioned is known for attracting strange people
E. The comedian's friend's mom is known for having bad eye-sight

*Figure 7.* Example of StandUp part of the RivaBench.

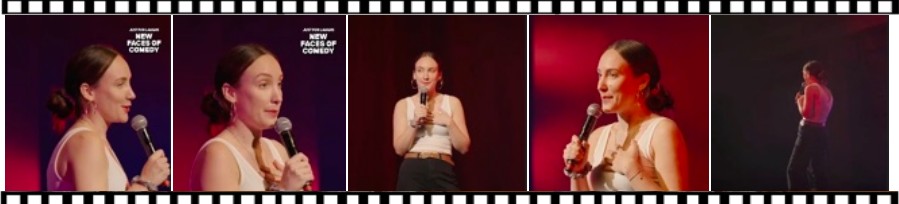

**Speech and audio:** I had so many, um, culture shocks since I moves here. For example as a foreigner I really hate tipping. [silence] But as a bartender I really hate foreigners. [audience laughed]

**Question**: Why did the audience laugh when she said "as a bartender I really hate foreigners"?

**Answer**: She uses a technique called "reversal" to make contrast

**Explain:** The audience laughs because the comedian employs a linguistic and comedic technique known as \"reversal\" or \"role inversion.\" By stating \"as a bartender I really hate foreigners,\" the comedian subverts the the statement as a foreigner I really hate tipping, which is a humorous and unexpected twist.

**Choices**:

A. The comedian was drinking alcohol on stage.

B. The comedian made a funny face while saying it

C. The comedian used a comedic technique called "reversal"

D. The audience agreed with her statement

E. The audience misunderstood her accent

*Figure 8.* Example of StandUp part of the RivaBench.

## C. Academic Data Examples

Two examples of the Academic part of RivaBench are shown in Fig. 9 and 10 respectively.

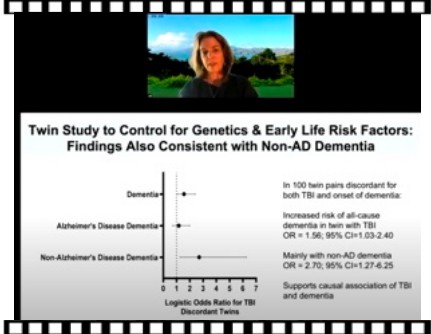

**Speech and audio:**…In which she's used a creative use of the twin pair kind of study design, and this is sort of interesting because you can. Because you can, because of their twins, who are discordant for Tbi and discordant for for onset of dementia. you control for early genetics, you control for early life exposures. And what Brenda has found is that, yes, indeed, Tbi is increased with with risk of dementia, about a 60 % increase exactly the. The same kind of effect size we've been finding, but it wasn't Alzheimer's. It seems like it's non Alzheimer's disease dementia. And I think we need to understand more. What is that? Is it vascular is? Is it Cte that's now met with aging? We don't really know…

**Question**: How does the twin study isolate TBI's impact on dementia risk?

**Answer**: By analyzing twins with inconsistent timing of onset of TBI and dementia.

**Explain:** The increased risk of \"all-cause dementia\" and \"non-Alzheimer's dementia\" mentioned in the slides suggests an association between TBI and an increased risk of dementia, particularly non-Alzheimer's type dementia...By analyzing inconsistent pairs, researchers can more accurately assess the association between TBI and dementia, thereby providing a scientific basis for prevention and intervention.

**Choices**:

A. By analyzing twins with inconsistent timing of onset of TBI and dementia.

B. By excluding early life exposures from the study.

C. By observing the progression of dementia in twins without TBI.

D. By ensuring differences in dementia risk are due to TBI.

E. By focusing only on Alzheimer's dementia outcomes.

*Figure 9.* Example of Academic part of the RivaBench.

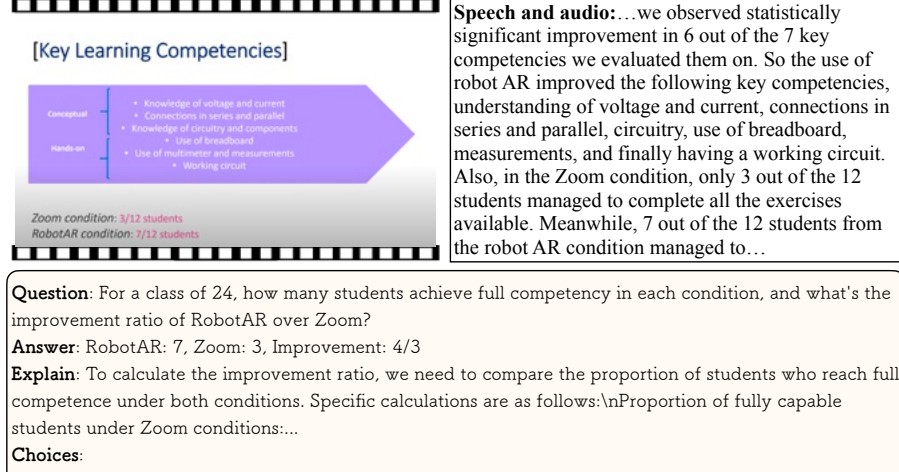

**Speech and audio:**…we observed statistically significant improvement in 6 out of the 7 key competencies we evaluated them on. So the use of robot AR improved the following key competencies, understanding of voltage and current, connections in series and parallel, circuitry, use of breadboard, measurements, and finally having a working circuit. Also, in the Zoom condition, only 3 out of the 12 students managed to complete all the exercises available. Meanwhile, 7 out of the 12 students from the robot AR condition managed to…

**Question**: For a class of 24, how many students achieve full competency in each condition, and what's the improvement ratio of RobotAR over Zoom?

**Answer**: RobotAR: 7, Zoom: 3, Improvement: 4/3

**Explain**: To calculate the improvement ratio, we need to compare the proportion of students who reach full competence under both conditions. Specific calculations are as follows:\nProportion of fully capable students under Zoom conditions:...

**Choices**:

A. RobotAR: 14, Zoom: 6, Improvement: 7/6.

B. RobotAR: 12, Zoom: 4, Improvement: 3/2.

C. RobotAR: 16, Zoom: 8, Improvement: 2/1.

D. RobotAR: 10, Zoom: 5, Improvement: 2/1.

E. RobotAR: 7, Zoom: 3, Improvement: 4/3

*Figure 10.* Example of Academic part of the RivaBench.

## D. Synthetic Video Detection Data Examples

Two synthetic video examples in the SynthDec partition of RivaBench are shown in Fig. 11 and 12 respectively.

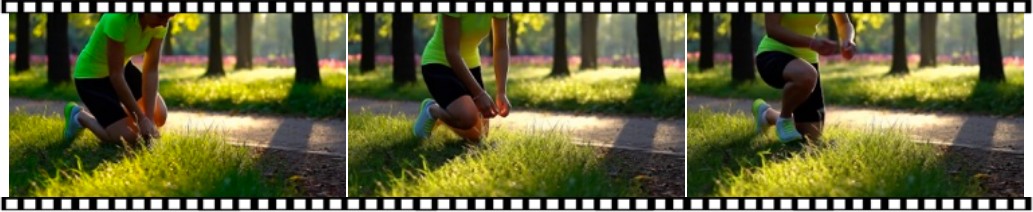

*Figure 11.* Example video clip of the SynthDec part of RivaBench.

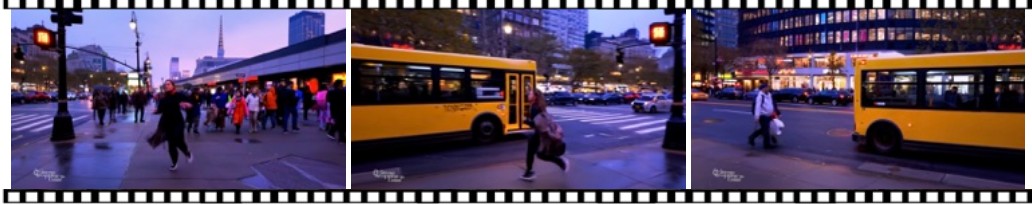

*Figure 12.* Example video clip of the SynthDec part of RivaBench.

## E. Prompt Templates

Prompt templates for video-SALMONN-o1 are shown in Table 5. When trained with the reasoning prompt template, the same template is used during inference. Changing templates may cause small perturbations in performance.

*Table 5.* Prompt used for different types of tasks.

| Type | Prompt content |
| --- | --- |
| Direct answer | <VIDEO>Select the best answer to the following question based on the video. Respond with only the letter of the correct option.
{Question}
Choose from: A. {Option A}, B, {Option B}... |
| Reasoning | <VIDEO> Question:
{Question}
Choose from: A. {Option A}, B, {Option B}...
Answer the question step by step. Output each thinking step. Mark the end of each step with <end_of_step> token. |
| SynthDec | An AI-generated video contains unnatural distorted things, such as distorted hands or faces. Is the given video AI generated? Answer YES or NO. Answer step by step and output each step clearly. |

# F. Case Studies: Solution with Reasoning Examples

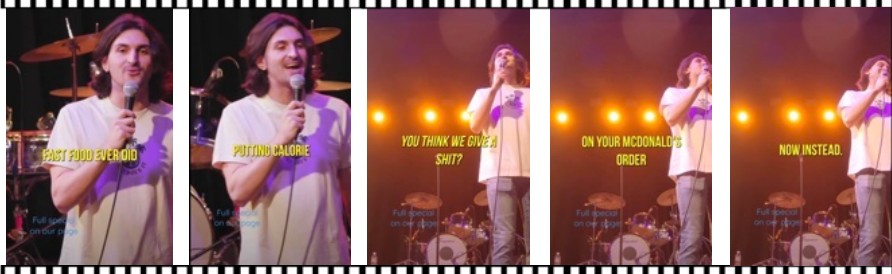

**Question:** What does the speaker imply by saying he 'didn't need to know that' at the end of the video?\nChoose from:\nA. He regrets ordering so much food.\nB. When you want to eat, just eat and don't prioritize calories.\nC. He doesn't like knowing calorie counts of food.\nD. He feels guilty about his food choices.\nE. The calorie count was surprisingly low.\nAnswer the question step by step. Output each thinking step.

**video-SALMONN-o1 SFT:**
1. The speaker is making a joke about how fast food restaurants started putting calorie counts on their menus.
2. He argues that this is a bad thing because it makes people feel guilty about eating large portions of unhealthy food.
3. He then says, 'I didn't need to know that,' implying that he doesn't like being reminded of the calorie count and feels like it's an unnecessary piece of information.
4. Therefore, the answer is **C. He doesn't like knowing calorie counts of food**. (Misinterpretation)

**video-SALMONN-o1 Process DPO:**
1. The The speaker is making a joke about how fast food restaurants use calorie counts to manipulate customers into buying more food.
2. He says he didn't need to know that because he already knew he was going to eat a lot of calories.
3. His tone and delivery suggest he's not actually upset about the calorie count, but rather finds it humorous that he doesn't need to be told how unhealthy his choices are. (Combining audio-visual information)
4. Therefore, the speaker implies that when you want to eat, just eat and don't prioritize calories.
5. The answer is B

*Figure 13.* Example video and solutions from the StandUp test set.

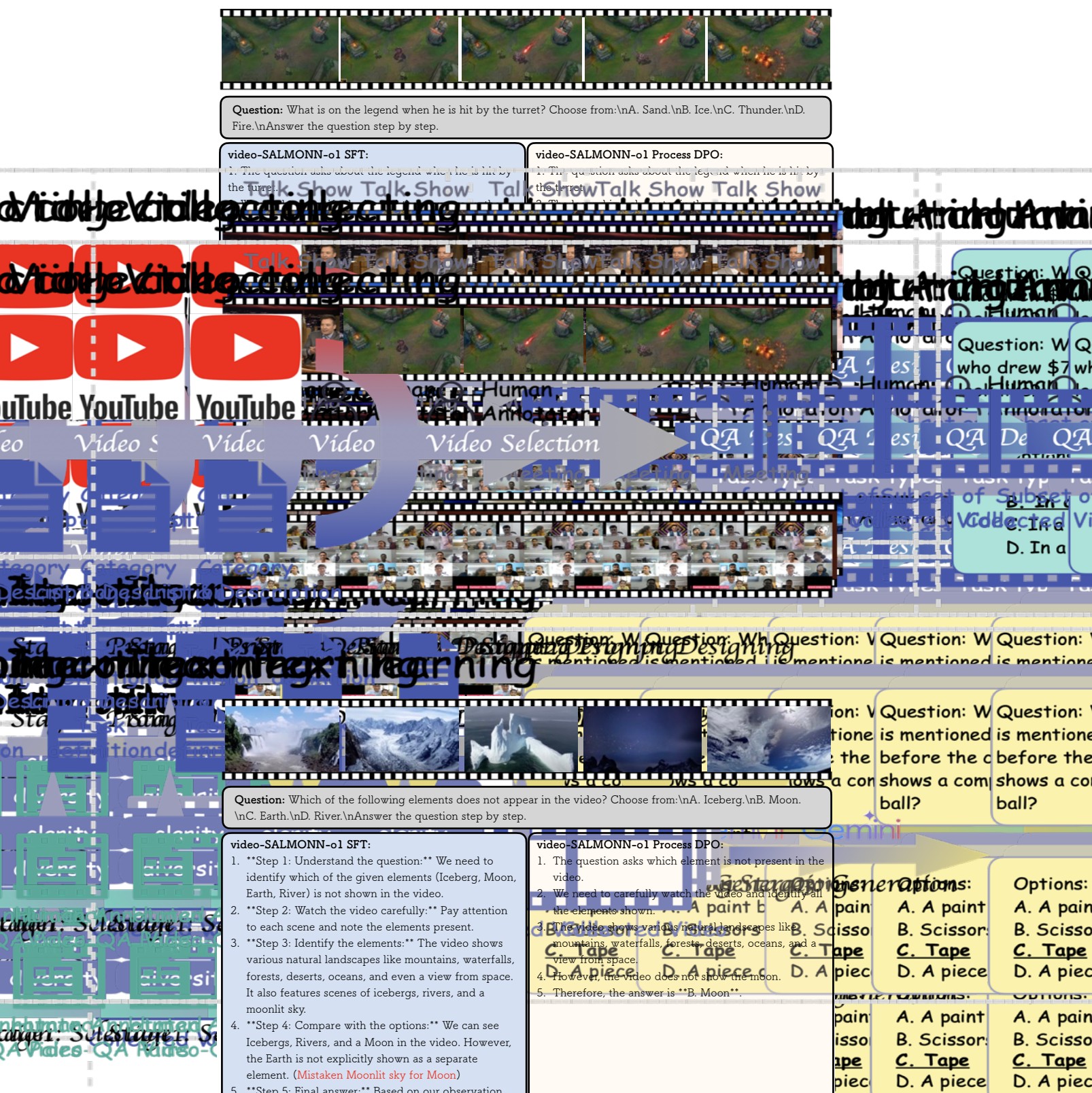

Figure 15. Example video and solutions from videoMME test set.

## G. Case Studies: Zero-shot Synthetic Video Detection

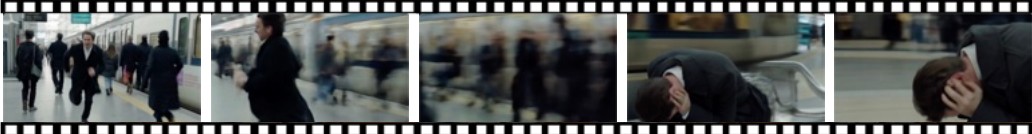

**Question:** An AI-generated video contains unnatural distorted things, such as distorted hands or faces. Is the given video AI generated? Answer YES or NO. Answer step by step and output each step clearly.

**LLaVA-OneVision:** No. The video is real.

**video-SALMONN-o1:**
The video shows a man running towards a train, then collapsing in pain.
The background is blurred and the colors are distorted, suggesting motion blur.
AI-generated videos often use techniques like motion blur to create a sense of speed or urgency.
This can sometimes result in unnatural distortions in the image.
In this case, the blurring and distortion are more pronounced than what would be expected from natural camera shake or movement.
The hands of the man appear particularly distorted, which is not typical for real-life footage.
Therefore, based on the visual cues of the video, it is likely that the video is AI generated.
So the answer is YES.

**GPT-4o:**
Let's analyze the video step by step:
**Clothing and Movement**: The person in the video is wearing a coat and is running. The movement appears natural, and the coat flows as expected with the motion. There are no visible distortions in the clothing or the movement of the person.
**Background and Environment**: The background shows a train station with people and a train. The environment looks consistent and realistic, with no obvious distortions or anomalies in the structure of the train, platform, or other people.
**Hands and Body Parts**: The hands and other visible body parts of the person appear normal and proportionate. There are no signs of unnatural distortions, such as extra fingers or oddly shaped limbs, which are common indicators of AI-generated content.
**Color and Lighting**: The color and lighting in the video appear consistent with a real-world setting. There are no unusual color shifts or lighting inconsistencies that would suggest AI generation.
**Conclusion**: NO, the video is not AI-generated..

**Gemini-1.5-pro:**
Here's how to figure out if the video is AI-generated:
**Examine the man's face while he's running:** Does it look unusually smooth, blurry, or distorted in any way?
**Pay close attention to the fall:** Do the movements look natural, or are they jerky and unrealistic?**Based on these observations, the answer is likely NO.**
The video doesn't exhibit the classic signs of AI-generated content. The man's face, while a bit obscured by the speed, doesn't show the telltale signs of AI distortion. Additionally, the fall, though quick, appears to have natural movement.
The final answer is: NO

*Figure 16.* Example output from video-SALMONN-o1, GPT-4o and Gemini-1.5-pro for synthetic video detection.

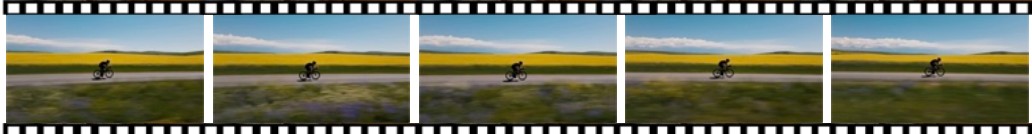

**Question:** An AI-generated video contains unnatural distorted things, such as distorted hands or faces. Is the given video AI generated? Answer YES or NO. Answer step by step and output each step clearly.

**LLaVA-OneVision:** NO.

**video-SALMONN-o1:**
The video shows a cyclist riding on a road with a field of flowers in the background.
The colors are vibrant and the scene is smooth, suggesting it might be AI generated.
Upon closer inspection, we can see that the cyclist's shadow is perfectly aligned with the cyclist, which is unusual for a moving object. This suggests a lack of natural motion blur or distortion, a common characteristic of AI-generated videos.
Therefore, based on the smoothness of the scene and the unnatural alignment of the cyclist's shadow, it is highly likely that the video is AI generated.
So the answer is YES.

**GPT-4o:** 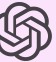
Here are the steps to follow:
**Subject Examination**: The cyclist appears to have a natural posture and movement. There are no visible distortions in the limbs or face that suggest AI generation
**Background Analysis**: The landscape, including the road, fields, and sky, appears consistent and natural. There are no obvious texture or perspective issues.
**Motion Artifacts**: The motion of the cyclist and the background blur due to speed appear natural and consistent with real-life video capture.
**Color Anomalies**: The colors in the video, including the yellow fields and blue sky, appear natural and consistent without any abrupt or unnatural transitions
**Overall Realism**: The video maintains a high level of realism throughout, with no elements that seem artificial or out of place.

**Gemini-1.5-pro:**
**Examine the cyclist's form:** Does the body shape look natural throughout the pedaling motion?  Are there any points where the limbs look distorted or bend in odd ways?
**Check the bike:**  Does the bike's movement seem realistic? Do the wheels spin smoothly?
Look for unusual artifacts: AI-generated content can sometimes have strange textures or patterns, particularly in areas of motion blur.
**Without the actual video, it's impossible to give a definitive YES or NO answer.**
**However**, if you observed any of the issues mentioned above while watching the video, it would be a strong indicator of AI generation.
You haven't provided me with the analysis of the video, so I can't give you a final answer of YES or NO.

*Figure 17.* Example output from video-SALMONN-o1, GPT-4o and Gemini-1.5-pro for synthetic video detection.

## H. Examples of Contrastive Step Selection Process

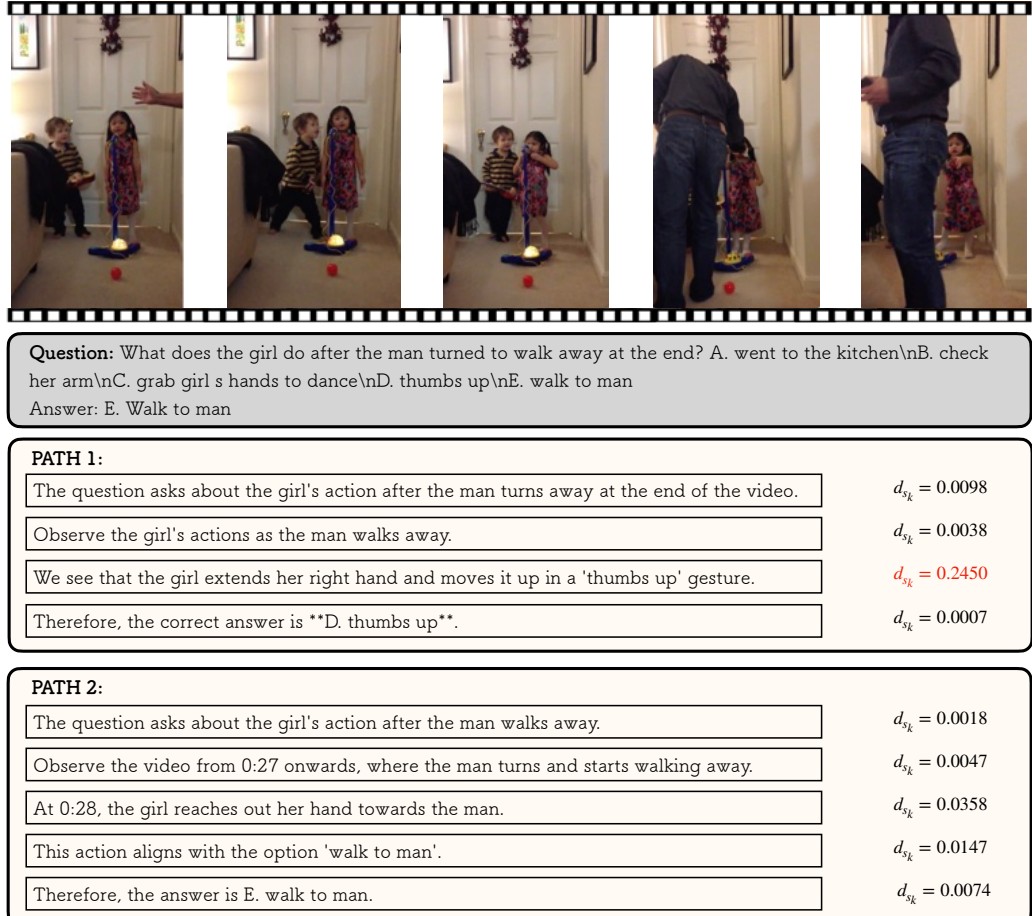

*Figure 18.* Example of the contrastive step selection process where two sampled paths are shown and the scores $d_{s_k}$ are given for each reasoning steps. The 3rd step in the first solution is wrong due to visual hallucination, and as a result, a very high score is assigned to that step and that step will be used to perform rollout.

## I. Additional Results on VideoHallucer

| Model | Object relation | Temperal | Semantic detail | Factual | Non-factual | Overall |
|---|---|---|---|---|---|---|
| Gemini-1.5-pro | 52.0 | 18.5 | 53.5 | 16.5 | 48.5 | 37.8 |
| video-SALMONN-o1 | 63.4 | 56.4 | 16.0 | 43.0 | 55.6 | 46.2 |

*Table 6.* video-SALMONN-o1 performance compared against Gemini-1.5-pro on VideoHallucer (Wang et al., 2024c) and report the overall accuracy (when the entire pair is correct) for each category.

