# OpenReview forum: "video-SALMONN-o1: Reasoning-enhanced Audio-visual Large Language Model"
_ICML.cc/2025/Conference — ICML 2025 poster_

### Official Review · Reviewer_dh9B · 2025-03-13

**Overall Recommendation:** 4

**Summary:**

This paper introduces video-SALMONN-o1, the first open-source reasoning-enhanced audio-visual LLM designed to address the underexplored challenge of general video understanding, which requires complex multimodal (audio-visual-text) reasoning. Current reasoning-optimized LLMs focus narrowly on mathematical/textual tasks or image inputs, limiting their applicability to real-world video scenarios like academic presentations or synthetic video detection. The authors propose a two-pronged approach: (1) a reasoning-intensive dataset with step-by-step solutions for supervised fine-tuning, and (2) process Direct Preference Optimization (pDPO), a novel training method that optimizes reasoning paths via contrastive step selection, eliminating the need for external reward models. They also introduce RivaBench, a benchmark with 4,000+ expert-curated QA pairs spanning diverse video contexts. Results show 3–8% accuracy gains over visual baselines (e.g., LLaVA-OneVision), 6–8% improvement from pDPO on RivaBench, and novel zero-shot synthetic video detection capabilities.

## update after rebuttal

Thanks for response. My concerns have all been solved.

**Claims And Evidence:**

Yes, they are well supported.

**Essential References Not Discussed:**

N/A

**Experimental Designs Or Analyses:**

Yes, I checked the experimental designs and analyses in Sec. 6.

**Methods And Evaluation Criteria:**

Yes, they do.

**Other Comments Or Suggestions:**

N/A

**Other Strengths And Weaknesses:**

## Strengths

- The technical contribution is solid, and the conducted experiments are convincing.
- The provided experimental results clearly show the effectiveness of the proposed method.

## Weaknesses

- Unclear inference mechanics: The paper lacks details about the inference pipeline. Please elaborate more on this. For example:
  - How many reasoning paths are generated per video-question pair during testing?
  - Is the reward model (or pDPO’s contrastive step selection) applied at inference time, and if so, how does this impact latency?

- Limited generalizability validation: While focused on audio-visual tasks, the proposed pDPO and reasoning dataset appear applicable to vision-only LLMs (e.g., LLaVA-OneVision). Testing these components on other architectures would strengthen claims about methodological universality.

**Questions For Authors:**

N/A

**Relation To Broader Scientific Literature:**

The reasoning-enhanced training strategy can be applied to broad video tasks.

**Theoretical Claims:**

N/A

---

> ### Author Rebuttal · Authors · 2025-03-31
>
> We deeply appreciate Reviewer dh9B for the positive comments and acknowledgement of our contribution. We would like to address the following questions:
> 1. __Inference mechanics__:
>   - We use greedy decoding during inference.
>   - We use contrastive step selection only to construct training preference pairs for pDPO. Hence no impact on latency.
> 2. __Generalizability validation__:
>   - While focusing on audio-visual tasks, we also present results on visual only tasks including NeXT-QA and SynthDec partition throughout the paper (Table 2-4).

---

> > ### Comment · Reviewer_dh9B · 2025-04-02
> >
> > Thanks for response. My concerns have all been solved.

---

### Official Review · Reviewer_3hnn · 2025-03-13

**Overall Recommendation:** 1

**Summary:**

The paper introduces video-SALMONN-o1, an open-source audio-visual LLM enhanced for general video understanding. It proposes pDPO for reasoning optimization and RivaBench, a new benchmark. The model shows improved accuracy over baselines and zero-shot synthetic video detection capabilities. However, the benchmark's reliance on LLM-generated data raises concerns about accuracy and bias.

**Claims And Evidence:**

The claims in the submission lack clear and convincing evidence. The benchmark results may be unreliable due to potential hallucinations and biases in the large models used to create them. Additionally, the authors fail to compare their model with true SOTA models like Gemini 1.5 Pro[3], Qwen2-VL[1], and GLM-4V-Plus[2], undermining the validity of their claims about achieving SOTA performance.


- [1] Wang, Peng, et al. "Qwen2-vl: Enhancing vision-language model's perception of the world at any resolution." arXiv preprint arXiv:2409.12191 (2024).
- [2] Hong, Wenyi, et al. "Cogvlm2: Visual language models for image and video understanding." arXiv preprint arXiv:2408.16500 (2024).
- [3] Team, Gemini, et al. "Gemini: a family of highly capable multimodal models." arXiv preprint arXiv:2312.11805 (2023).

**Essential References Not Discussed:**

No.

**Experimental Designs Or Analyses:**

I checked the experimental designs and found issues, particularly with the benchmark construction. The RivaBench was generated using large language models, which can introduce hallucinations and biases, potentially compromising its validity. Additionally, the authors did not compare their model against true SOTA models like Gemini 1.5 Pro, Qwen2-VL, or GLM-4V-Plus, which raises concerns about the robustness of their claims.

**Methods And Evaluation Criteria:**

The proposed methods, including the reasoning-enhanced audio-visual LLM and pDPO, are relevant for general video understanding. However, the evaluation benchmark RivaBench is generated by LLMs, which may introduce biases and inaccuracies due to hallucinations. Overall, while the methods are sensible, the evaluation criteria have significant limitations.

**Other Comments Or Suggestions:**

1. Include supplementary material to support claims and provide detailed methodology.
2. Validate benchmark results with human annotations to mitigate model biases and hallucinations.
3. Compare with true SOTA models (e.g., Gemini 1.5 Pro, Qwen2-VL, GLM-4V-Plus) for a fair evaluation.

**Other Strengths And Weaknesses:**

Strengths: The paper introduces video-SALMONN-o1, a novel open-source audio-visual LLM with enhanced reasoning abilities, and proposes pDPO for efficient step-level reward modeling. It also introduces RivaBench, a reasoning-intensive benchmark.

Weaknesses: The benchmark relies on LLM-generated data, which may introduce biases and hallucinations. The paper lacks comparisons with true SOTA models like Gemini 1.5 Pro and Qwen2-VL, potentially overstating its achievements. Supplementary materials are absent, limiting reproducibility.

**Questions For Authors:**

How do you address potential biases and hallucinations in the benchmark data generated by LLMs?

**Relation To Broader Scientific Literature:**

No broader impact.

**Theoretical Claims:**

The paper does not present any formal theoretical proof. It focuses on empirical results and methodological contributions, such as the introduction of video-SALMONN-o1 and pDPO. Therefore, there are no proofs to verify.

---

> ### Author Rebuttal · Authors · 2025-03-31
>
> We thank the reviewer for the comments and would like to resolve concerns and misunderstandings as follows:
> 1. __Regarding the reliability and bias of the benchmark__:
>   - As stated in section 5 paragraphs 2 and 3, we __always__ use human annotators to generate questions and answers, and __always__ use human annotators to check and validate the questions and answers.
>   - We employed over 50 __expert__ annotators to perform this task. This way, we ensure diversity and minimize bias and hallucination to a level that is __above average human beings__.
>   - Please refer to the "__RivaBench Demo Samples__" section of the demo page in the paper for concrete examples of human-created questions:
>     - One medical example requires professional medical knowledge. The explanation reflects the annotator's knowledge.
>     - One math example requires the ability to understand parabolic partial differential equations. See the detailed step-by-step derivation provided by the annotator.
>     - All samples are accompanied with such expert explanations and will be released upon acceptance.
>
> 2. __Comparisons with other SOTA methods__:
>   - We have compared LLaVA-OneVision in Table 2 as a strong open-source video understanding model baseline.
>   - We included the performance of Qwen2-VL as follows:
> | Models | VideoMME | NeXT-QA | Academic | StandUp | SynthDec |
> | -------- | ------- | ------- | ------- | ------- | ------- |
> | Qwen2-VL | 62.9 | 80.2 | 48.2 | 71.6 | 0.0 |
> | video-SALMONN-o1| __65.6__ | __82.3__ | __48.3__ | __76.7__ | __17.8__ |
>   - video-SALMONN-o1 achieved superior performance on all the benchmarks. The improvement on the academic partition is smaller because this data needs __less__ audio-visual understanding but __more__ mathematical reasoning. However, pDPO focuses more on optimizing audio-visual understanding in the reasoning process.
>   - We have also included the results for Gemini-1.5-pro in Table 2. Note that Gemini-1.5-pro is a proprietary LLM with unknown (potentially much larger) model sizes and unknown inference mechanism, and hence no direct comparison should be drawn with it. Given that, our model __still outperforms__ Gemini-1.5-pro on NeXT-QA and StandUp.

---

### Official Review · Reviewer_cRid · 2025-03-14

**Overall Recommendation:** 4

**Summary:**

This paper introduces video-SALMONN-o1, an open-source reasoning-enhanced audio-visual LLM designed for general video understanding tasks. The authors claim that existing reasoning models are merely focusing on either math problems or visual graphical inputs, without sufficient attention on general audio-video understanding. To fill this gap, they proposed video-SALMONN-o1 to tackle complex reasoning problems in the field of audio-video understanding. The main contributions include 1) a reasoning-intensive dataset featuring challenging audio-visual questions with step-by-step solutions, 2) a novel process direct preference optimization (pDPO) algorithm, and 3) RivaBench, the first reasoning-intensive video understanding benchmark, with over high-quality, expert-curated question-answer pairs across diverse scenarios. Extensive experiments demonstrate the effectiveness of the proposed scheme.

**Claims And Evidence:**

The key claim of this paper is about the state of existing reasoning-enhanced MLLM. The authors claim that existing models merely focus on narrow scenarios such as math, coding, or simple visual graphic inputs, while the exploration in general video understanding is missing. The above claim is correct and reasonable, as the community is still exploring the proper reasoning paradigms for video understanding.

**Essential References Not Discussed:**

N/A

**Experimental Designs Or Analyses:**

The experiments were conducted on both public benchmarks for general video understanding and the proposed RivaBench. Results on these benchmarks can well-demonstrate the effectiveness and the significance of the proposed scheme.

**Methods And Evaluation Criteria:**

Yes. The proposed method for reasoning-enhanced audio-visual understanding is reasonable. The interleaved synchronization design can better handle the multimodal audio-visual data. The evaluation metric for RivaBench is accuracy (MCQ or Yes/No), which is a common standard in evaluating MLLMs.

**Other Comments Or Suggestions:**

N/A

**Other Strengths And Weaknesses:**

Generally, this is a good paper about LLM-based & reasoning-enhanced video understanding. Extending existing paradigms to audio/video understanding is a non-trivial setting. The proposed video-SALMONN-o1 model can well-handle such scenarios.

**Questions For Authors:**

The authors are encouraged to open-source the code and data mentioned in the submission, which would be a good contribution to the community.

**Relation To Broader Scientific Literature:**

The key contribution of this paper is to generalize the reasoning capabilities of LLMs/MLLMs to video understanding, especially in the case when both audio and visual information are used. It diverses this work from existing publications.

**Theoretical Claims:**

The paper does not contain any proofs or theoretical claims.

---

> ### Author Rebuttal · Authors · 2025-03-31
>
> We deeply thank you for acknowledging our effort and contribution!

---

> > ### Comment · Reviewer_cRid · 2025-04-02
> >
> > Thanks for the response from the authors. I'm keeping my original rating.

---

### Official Review · Reviewer_BoZJ · 2025-03-18

**Overall Recommendation:** 3

**Summary:**

In this work, an RL-based optimization and reasoning-aware framework is proposed for training a large audio-video multi-modal model called Video-SALMONN-o1. This work emphasizes that significant effort has been invested in improving the mathematical and visual graphical inputs from the RL perspective, leading to the introduction of RL-based approaches for video-understanding models. To achieve this, a new step-by-step reasoning-based video-LLM training dataset has been proposed. This dataset is used to perform reasoning-based Supervised Fine-Tuning (SFT) and RL fine-tuning stages. For the second stage, a combination of PPRM and process DPO objectives has been formulated to improve both solution-level and fine-grained step-level preference training. To enhance training efficiency in the pDPO method, contrastive step selection is utilized, which applies optimization to steps that are most sensitive to video-based changes. The proposed model is trained using multiple stages, including SFT and the RL stage. To further benchmark audio-video models, a new benchmark called RivaBench is introduced, which effectively evaluates the model's reasoning and audio-encoding capabilities. After complete training, Video-SALMONN-o1 is evaluated on several benchmarks, and ablation studies are performed to justify the various optimization design choices proposed in the framework.

## update after rebuttal

Dear Authors,

Thank you for providing a rebuttal response. The majority of my concerns have been sufficiently addressed. I strongly recommend the authors incorporate all discussions and additional experiments presented in the rebuttal period into the main manuscript.

I will keep my original rating.

**Claims And Evidence:**

Yes, the main claims made in the submission are supported by extensive empirical benchmarking results and qualitative results (mostly in the supplementary part). The effectiveness of proposed RL-based framework is effective compared to other video-based foundation models.

**Essential References Not Discussed:**

RL-based literature for vision-based LLM models is not discussed in the manuscript. It is important that this work provides a related work section for RL-based methods (e.g., [2], [3]) for image domains and contrasts it with the proposed techniques.


[2] RLAIF-V: Aligning MLLMs through Open-Source AI Feedback for Super GPT-4V Trustworthiness
[3] Direct Preference Optimization of Video Large Multimodal Models from Language Model Reward

**Experimental Designs Or Analyses:**

I believe there are several aspects of the proposed dataset which are not clearly discussed in the manuscript.

**1) Validating the design choices in preparing SFT Dataset:**
i) It is unclear what are the initial data source for preparing the reasoning-based version of it. One of the main contributions of this work is to propose a RL/reasoning-friendly dataset. Unfortunately, details on that front are majorly missing. Additionally, discussion and statistics must be provided for various aspects such as QA count, number of videos, video-category distribution, etc.

ii) As mentioned in lines 138-140 (right column), what concrete filtering steps are employed to reject lower-quality QA samples?

iii) In Tab. 1, the duration for the Academic category has a strange standard deviation of 66.1 seconds, which is larger than the mean itself.

**2) Missing details regarding the pDPO dataset**:
i) The statistics and curation process of the proposed pDPO dataset has been discussed very late in the manuscript. A dedicated discussion should be added in the manuscript upfront in the initial sections of the manuscript. Instead of too much text, a visual diagram is recommended which outlines the major steps in the pDPO dataset generation process.

**3) Model Design Choices:**
i) It is unclear exactly what part of the model is frozen and what is trainable in each stage of the training. For example, it seems like most model components are frozen as written in lines 268-270 (right column).

**4) RivaBench:** The manuscript has provided a few video examples of the proposed evaluation benchmark. It can be noticed that in most examples, the speech lyrics are directly rendered on the video. This could cause models to overlook the audio tokens and directly read the subtitles from the visual inputs which is a shortcut. How do the authors aim to resolve this issue?

**5) Evaluations and Experiments:**

i) In my understanding, the proposed Video-LMM is based on video-SALMONN 2 model, but unfortunately, a performance comparison with this model is not performed.


**6) Training compute and efficiency comparison**: As the proposed framework utilizes the RL-based pDPO technique, how much extra compute and training cost is incurred compared to the baseline model? A clear compute cost analysis should be provided for the proposed framework.

**Methods And Evaluation Criteria:**

Overall, the evaluation criteria make sense for the problem at hand. However, explicitly checking the hallucination performance of the resultant model is not evaluated. There are test-only benchmarks such as [1] where the proposed method should be tested to validate the proposed framework in terms of its ability to reduce hallucinations in video-related tasks.

[1] VideoHallucer: Evaluating Intrinsic and Extrinsic Hallucinations in Large Video-Language Models

**Other Comments Or Suggestions:**

Minor:
In line 133, there is a typo error.

**Other Strengths And Weaknesses:**

Overall, the proposed training and evaluation datasets, alongside the RL-based optimization framework are motivating and useful for the research community. However, the overall presentation style of the manuscript is not good, too generic, and many important details are missing in the paper. It is strongly recommended to address all the comments mentioned in the earlier sections of the manuscript.

**Questions For Authors:**

Please refer to the above parts for questions.

**Relation To Broader Scientific Literature:**

This paper aims to advance the audio-video-language reasoning abilities of foundation models using reasoning and RL based training frameworks.

**Theoretical Claims:**

No explicit proofs are proposed in the manuscript.

---

> ### Author Rebuttal · Authors · 2025-03-31
>
> We sincerely appreciate the detailed and constructive reviews provided by Reviewer BoZJ. We would like to address the concerns and suggestions as follows:
> 1. We follow the evaluation described in VideoHallucer [1] and report the overall accuracy (when the entire pair is correct) for each category as follows:
> | Model    | Object relation | Temperal | Semantic detail | Factual | Non-factual | Overall |
> | -------- | ------- | ------- | ------- | ------- | ------- | ------- |
> | Gemini-1.5-pro [1] | 52.0 | 18.5 | __53.5__ | 16.5 | 48.5 | 37.8 |
> | video-SALMONN-o1 | __63.4__ | __56.4__ | 16.0 | __43.0__ | __55.6__ | __46.2__ |
>
> [1] VideoHallucer: Evaluating Intrinsic and Extrinsic Hallucinations in Large Video-Language Models
>
> 2. We appreciate the suggestions and hence provide the following descriptions on how we will improve the presentation in the revised version.
>   - __Validating the design choices__:
>     - __The SFT set__: contains 20k videos with 30k QA pairs. Mean duration 97.3s and maximum duration 256.1s. The majority of videos are randomly downloaded general YouTube videos, with 5k videos focusing on talks and presentations. We provide metadata files in the github repo.
>     - __How to filter__: We take the video description, question and answer and feed that into GPT-4o and ask if this question requires complicated logical reasoning. Keep the QA pair if the answer is Yes, and discard if No.
>     - __The large standard deviation of Academic category__: It contains math lectures which may last over 20 minutes, and short conference presentations that only last 1 or 2 minutes.
>   - __pDPO dataset__:  The description is provided in Section 6.2 from line 307. The above reasoning-intensive SFT data is used for pDPO. No additional data is generated for pDPO.
>   - __Model Design Choices__: The visual and audio encoders are frozen. The backbone LLM is trained with LoRA (rank=64) and the modality aligner (24M parameters) is trained.
>   - __Subtitle shortcut in RivaBench__:
>     - This only appears in 50% of the StandUp partition and none in the Academic partition.
>     - Even with subtitles, the frame rate of the video is low enough, such that the subtitles are often incomplete as seen by the visual encoder.
>     - The model has to leverage both audio and visual information, and this is reflected by the fact that GPT-4o (visual only) is worse than Gemini-1.5-pro (audio-visual).
>   - __Regarding comparison to video-SALMONN-2__: video-SALMONN-2 is not open-source yet. The proposed model only resembles the model structure and some opensource data. We do not have or use any model parameters from video-SALMONN-2.
>   - __Computational Cost__: The two additional computations are (i) sample generation and (ii) training the model with pDPO. For (i), we used 16 A100 GPUs for 48 hours to generate the number of samples described in lines 317-319. This is a one-off, and we do not need to generate them again. (ii) We use eight A100 GPUs for 24 hours to train pDPO (see lines 270-271). __Note that these are all training time costs, and it does not incur any additional test time costs at all.__
>   - __Open Source__: We promise to provide model checkpoint, training and inference code, as well as benchmark data upon acceptance of the paper.
>
> 3. We will also include the following reference [2, 3] in section 2 of our revised paper:\
> [2] RLAIF-V: Aligning MLLMs through Open-Source AI Feedback for Super GPT-4V Trustworthiness \
> [3] Direct Preference Optimization of Video Large Multimodal Models from Language Model Reward

---

### Decision · Program_Chairs · 2025-05-01

**Decision:**

Accept (poster)

**Comment:**

The manuscript received ratings of 4, 4, 3, and 1. Reviewers appreciated that the proposed training and evaluation datasets, alongside the RL-based optimization framework are motivating, useful for the research community, and the results demonstrate the effectiveness of the proposed scheme. Reviewers also raised several questions including, missing details regarding the pDPO dataset, model design choices, training compute and efficiency comparison, and the benchmark relies on LLM-generated data. The authors provided a detailed rebuttal to address reviewers concerns. Post-rebuttal, reviewer BoZJ mentioned that majority of the reviewer's concerns are sufficiently addressed. To address the concerns of reviewer 3hnn about the reliance on LLM-generated data, the authors also clarified that human annotators are always used to generate questions and answers and to check and validate the questions and answers. Further, the authors clarified that they employed over 50 expert annotators to perform this task. The AC believes this addresses the concerns of reviewer 3hnn, which were not acknowledged by the reviewer post-rebuttal. Given that the remaining three reviewers are generally positive and the authors addressed the main concern of the fourth reviewer regarding potential issues in the data (e.g., LLM-generated data), the AC agrees that the proposed approach has merits and will be interesting for the research community.